# VoxGenesis: Unsupervised Discovery of Latent Speaker Manifold for Speech Synthesis

## Abstract

Achieving nuanced and accurate emulation of human voice has been a longstanding goal in artificial intelligence. Although significant progress has been made in recent years, the mainstream of speech synthesis models still relies on supervised speaker modeling and explicit reference utterances. However, there are many aspects of human voice, such as emotion, intonation, and speaking style, for which it is hard to obtain accurate labels. In this paper, we propose VoxGenesis, a novel unsupervised speech synthesis framework that can discover a latent speaker manifold and meaningful voice editing directions without supervision. VoxGenesis is conceptually simple. Instead of mapping speech features to waveforms deterministically, VoxGenesis transforms a Gaussian distribution into speech distributions conditioned and aligned by semantic tokens. This forces the model to learn a speaker distribution disentangled from the semantic content. During the inference, sampling from the Gaussian distribution enables the creation of novel speakers with distinct characteristics. More importantly, the exploration of latent space uncovers human-interpretable directions associated with specific speaker characteristics such as gender attributes, pitch, tone, and emotion, allowing for voice editing by manipulating the latent codes along these identified directions. We conduct extensive experiments to evaluate the proposed VoxGenesis using both subjective and objective metrics, finding that it produces significantly more diverse and realistic speakers with distinct characteristics than the previous approaches. We also show that latent space manipulation produces consistent and human-identifiable effects that are not detrimental to the speech quality, which was not possible with previous approaches. Finally, we demonstrate that VoxGenesis can also be used in voice conversion and multi-speaker TTS, outperforming the state-of-the-art approaches. Audio samples of VoxGenesis can be found at: `https://bit.ly/VoxGenesis`.

## 1 Introduction

Deep generative models have revolutionized multiple fields, marked by several breakthroughs including the Generative Pretrained Transformer (GPT) (Brown et al., 2020), Generative Adversarial Network (GAN) (Goodfellow et al., 2014), Variational Autoencoder (VAE) (Kingma & Welling, 2014), and, more recently, Denoising Diffusion Models (DDPM) (Dhariwal & Nichol, 2021; Ho et al., 2020). These models can generate realistic images, participate in conversations with humans, and compose intricate programs. When utilized in speech synthesis, they are capable of producing speech that is virtually indistinguishable from human speech (Shen et al., 2018; Oord et al., 2016; Kim et al., 2021; Wang et al.; Tan et al., 2022). However, the success are primarily confined to replicating the voices of training or reference speakers. In contrast to image synthesis, where models can produce realistic and unseen scenes and faces, the majority of speech synthesis models are unable to generate new, unheard voices. We argue that this limitation predominantly stems from the design of the speaker encoders and neural vocoders. Typically, they function as deterministic modules (Polyak et al., 2021; Jia et al., 2018b; Kim et al., 2021; Qian et al., 2020), mapping speaker embeddings to the target waveforms.

Besides the obvious advantage of being able to generate new objects, generative models also permit the control over the generation process and allow for latent space manipulation to edit specific aspects of the generated objects without the necessity for attribute labels (Härkönen et al., 2020;

Voynov & Babenko, 2020a). This advantage is especially important in speech, where nuanced characteristics such as emotion, intonation, and speaker styles are hard to label. The incorporation of a speaker latent space could enable more sophisticated voice editing and customization, expanding the potential applications of speech synthesis substantially. However, learning the speaker distribution is not a straightforward task. This is due to the intrinsic complexity of speech signals where speaker-specific characteristics are entangled with the semantic content information. As such, we cannot directly fit a distribution over speech and expect the model to generate new speakers while maintaining control over the content information. The disentanglement of content from speaker features is a necessary first step (Hsu et al., 2017; Yadav et al., 2023; Qian et al., 2020; Lin et al., 2023). In (Stanton et al., 2022), the authors proposed TacoSpawn, a method that fits a Gaussian Mixture Model (GMM) over Tacotron2 speaker embeddings to learn a prior distribution over speakers. While TacoSpawn (Stanton et al., 2022) does offer the capability to generate novel speakers, it comes with its own set of limitations. Firstly, there is a separation in the parameterization of the speaker embedding table and the speaker generation model, which prevents the synthesis modules from fully benefiting from the generative approach. Secondly, in contrast to modern deep generative models, the mixture model in TacoSpawn is trained to maximize the likelihood of the speaker embeddings rather than the data likelihood, thereby limiting the representational capability of the generative model.

In this paper, we introduce VoxGenesis, an unsupervised generative model that learns a distribution over the voice manifold. At its core, VoxGenesis learns to transform a Gaussian distribution into a speech distribution conditioned on semantic tokens. This approach contrasts with conventional GAN vocoders such as Mel-GAN (Kumar et al., 2019), HIFI-GAN (Kong et al., 2020), and more recently SpeechResynthesis (Polyak et al., 2021), which learn a deterministic mapping between speech features and waveforms. Figure 1 illustrates the architectural differences between VoxGenesis and SpeechResynthesis. VoxGenesis introduces a mapping network that converts the isotropic Gaussian distribution into a non-isotropic one, enabling the control module (the yellow box) to identify major variances. It also features a shared embedding layer for the discriminator and employs semantic transformation matrices, facilitating semantic-specific transformations of speaker attributes. Furthermore, VoxGenesis sets itself apart from image generation GANs like Style-GAN or BigGAN by integrating a Gaussian constrained encoder into the framework. This inclusion not only stabilizes training but also enables the encoding of external speaker representations.

In summary, our contributions are as follows:

- We introduce a general framework for unsupervised voice generation by transforming Gaussian distribution to speech distribution.
- We demonstrate the potential for unsupervised editing of nuanced speaker attributes such as gender characteristics, pitch, tone, and emotions.
- We identify the implicit sampling process associated with using speaker embeddings for GANs and proposed a divergence term to constrain the speaker embeddings distributions. This allows the conventional speaker encoder to be incorporated as components of a generative model, thereby facilitating the encoding and subsequent modification of external speakers.

## 2 BACKGROUND

**Voice Conversion (VC) and Text-to-Speech (TTS).** The majority of VC and TTS models work in the speech feature domain, meaning that the model output are speech features such as a Mel-spectrogram (Qian et al., 2019; Kaneko et al., 2019; Shen et al., 2018; Ren et al., 2019). The primary distinction between VC and TTS is their approach to content representations. While VC strives to convert speech from one speaker to another without altering the content, TTS acquires content information from a text encoder. This encoder is trained on paired text-speech data and can utilize autoregressive modelling (Shen et al., 2018) or non-autoregressive modelling with external alignments, as seen in FastSpeech (Ren et al., 2019). Given the necessity for vocoders in both VC and TTS to invert the spectrogram, there has been a significant effort to improve them. This has led to the development of autoregressive, flow, GAN, and diffusion-based vocoders (Kong et al., 2020; Oord et al., 2016; Prenger et al., 2019; Chen et al., 2020). Recently, VITS (Kim et al., 2021), an end-to-end TTS model, has been introduced; it utilizes conditional variational autoencoders in

tandem with adversarial training to facilitate the direct conversion from text to waveform. Building on top of VITS, YourTTS (Casanova et al., 2022) caters to multilingual scenarios in low-resource languages by enhancing the input text with language embeddings. Beyond the standard GAN and VAE models, VoiceBox introduces flow-matching to produce speech when provided with an audio context and text (Le et al., 2023).

**Speaker Modeling in Speech Synthesis.** Speaker modeling stands as a crucial component in speech synthesis. The initial approach to speaker modeling involved the utilization of a speaker embedding table (Gibiansky et al., 2017). However, the scalability of this method becomes a concern with the increase in the number of speakers. Therefore, pretrained speaker encoders have been introduced into TTS systems to facilitate the transfer of learned speaker information to the synthesis modules (Jia et al., 2018a). The speaker embeddings can be combined with conventional speech features like MFCC or with self-supervised learned (SSL) speech units (Hsu et al., 2021; Schneider et al., 2019). A demonstration of integrating SSL speech units with speaker embeddings is presented in SpeechResynthesis (Polyak et al., 2021), where the authors have proposed a model that re-synthesizes speech utilizing SSL units and speaker embeddings. Besides the explicit utilization of speaker lookup tables and speaker embeddings, speaker information can also be incorporated implicitly. This can be achieved by training autoregressive models on residual vector quantized (RVQ) representations (Kumar et al., 2023; Défossez et al., 2022; Zeghidour et al., 2021), exemplified by VALL-E (Wang et al.), or through BERT-like masking prediction as in SoundStorm (Borsos et al., 2023).

**Speech Style Learning and Editing.** Speech conveys multifaceted information such as speaker identity, pitch, emotion, and intonation. Many elements are challenging to label, making unsupervised learning a popular approach for extracting such information. The concept of Style Tokens is introduced in (Wang et al., 2018), where a bank of global style tokens (GST) is learned jointly with Tacotron. The authors demonstrate that GST can be employed to manipulate speech speed and speaking style, independently of text content. In another development, SpeechSplit (Wang et al., 2018) achieves the decomposition of speech into timbre, pitch, and rhythm by implementing information bottlenecks. Consequently, style-transfer can be executed using the disentangled representations. However, the utilization of information bottlenecks can potentially result in deteriorated reconstruction quality. To address this issue, the authors in (Choi et al., 2021) propose NANSY, an analysis and synthesis framework that employs information perturbation to disentangle speech features. This method has been successfully applied in various applications, including voice conversion, pitch shift, and time-scale modification.

## 3 VoxGenesis

### 3.1 Learning Latent Speaker Distribution with GAN

Generative Adversarial Network (GAN) has been the de facto choice for vocoders since the advent of Mel-GAN and HiFi-GAN (Kumar et al., 2019; Kong et al., 2020). However, these GANs are predominantly utilized as spectrogram inverters, learning deterministic mappings from Mel-spectrogram or other speech features (Polyak et al., 2021) to waveforms. A notable limitation in these models is the lack of true "generation"; the vocoders learn to replicate the voices of the training or reference speakers rather than creating new voices. This stands in contrast to the application of GANs in computer vision, where they are primarily utilized to generate new faces and objects (Karras et al., 2019; Brock et al., 2018). The principal challenge in using GAN as a generative model arises due to the high semantic variations in speech. This makes transforming a Gaussian distribution to a speech distribution difficult without certain constraints. Recently, Self-Supervised Learned (SSL) speech units have emerged as effective tools for disentangling semantic information (Hsu et al., 2021; Baevski et al., 2020). This advancement motivates us to utilize these semantic tokens as conditions for GAN; consequently, a conditional GAN is employed to transform a Gaussian distribution, rather than mapping speech features to waveforms. Let's represent a semantic token sequence as $Y$ and an acoustic waveform sequence as $X$, originating from the empirical distribution $p_{\text{data}}(X)$. We aim to train a GAN to transform a standard Gaussian distribution $p(\mathbf{z})$ into speech distributions $p_{\text{data}}(X)$ conditioned on semantic tokens $Y$. This is done by solving the following min-max problem with a discriminator $D$ and a generator $G$:

$$\min_G \max_D V(D, G) = \mathbb{E}_{X \sim p_{\text{data}}(X)}[\log D(X \mid Y)] + \mathbb{E}_{\mathbf{z} \sim \mathcal{N}(0,I)}[\log(1 - D(G(\mathbf{z} \mid Y)))], \quad (1)$$

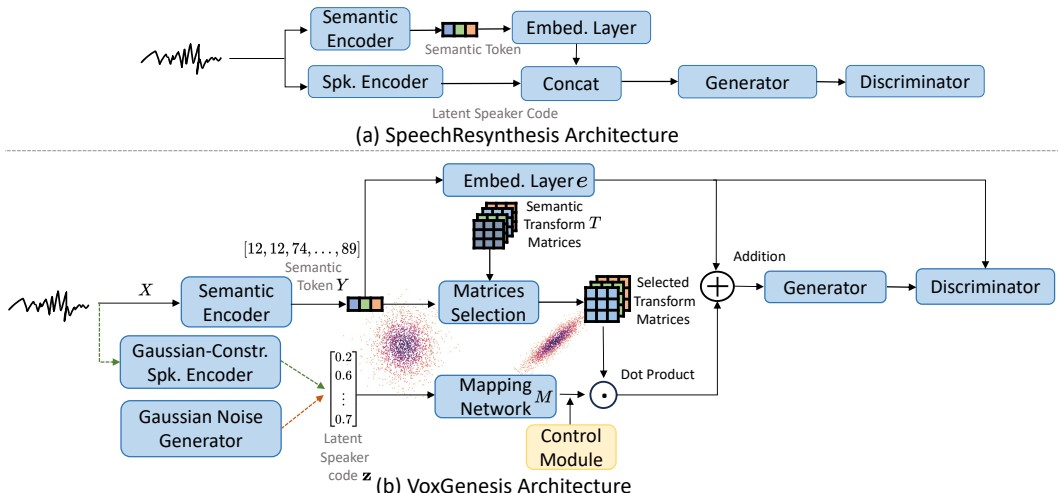

Figure 1: Illustration of (a) the SpeechResynthesis (excluding pitch module) and (b) our VoxGenesis. In contrast to SpeechResynthesis, which focuses on reconstructing the waveforms from semantic tokens and speaker embeddings, VoxGenesis is a deep *generative* model that learns to transform a Gaussian distribution to match the speech distribution conditioned on semantic tokens either using Gaussian noise as input (the green dashed line), or using embeddings from a Gaussian-constrained speaker encoder (pink dashed line).

There is no one-to-one correspondence between the latent code and the generated waveform anymore. Therefore, training the network with the assistance of Mel-spectrogram loss, as proposed in (Kong et al., 2020), is no longer feasible here. Instead, the model is trained to minimize discrepancies only at the distribution level, as opposed to relying on point-wise loss.

Regarding the generator's design, Figure 1(b) highlights three modules that are vital for learning:
**Shared Embedding Layer** $e$: Both the generator and the discriminator leverage a shared embedding layer $e$. It is crucial, in the absence of Mel-spectrogram loss, for the discriminator to receive semantic tokens; otherwise, the generator could deceive the discriminator with intelligible speech.
**Mapping Network** $M$: A mapping network $M$ is integrated, consisting of seven feedforward layers, to transform the latent code prior to the deconvolution layers. Drawing inspiration from Style-GAN (Karras et al., 2019), this enables the generation of more representative latent codes and a non-isotropic distribution, the output of which will be utilized by the control module, discussed in later section. **Semantic Conditioned Transformation** $T$: Rather than indiscriminately adding the latent codes to each semantic token embedding, we conditionally transform the latent code based on the semantic information. This enables semantic-specific transformations of speaker attributes. Specifically, the generator comprises a deep deconvolution network $f$, a semantic conditioned feed-forward network $T$, a shared embedding layer $e$, and a latent code transform network $M$. The equation for the generator, conditioned on semantic tokens, is represented as:

$$G(\mathbf{z}|Y) = f\left(T\big(M(\mathbf{z}), Y\big) + e(Y)\right). \tag{2}$$

**Ancestral Sampling for GAN.** Transforming random noise to speech distribution has its disadvantages, one of which is the inability to use specific speakers' voices post-training due to the absence of an encoder to encode external speakers. Another notable challenge is the well-known "mode collapse," an issue often mitigated in most GAN vocoders due to the stabilizing effect of the Mel-spectrogram loss during training. To overcome these challenges, we introduce a probabilistic encoder capable of encoding speaker representation using posterior inference, $p_\theta(\mathbf{z}|X)$, while maintaining the marginal distribution as a standard Gaussian distribution, $p(\mathbf{z})$. The neural factor analysis (NFA) (Lin et al., 2023) is one of such models. Here we assume the NFA encoder is pre-trained. During GAN training, ancestral sampling is used; initially, samples are drawn from the empirical distribution, $p_{\text{data}}(X)$, followed by sampling from the posterior distribution , $p_\theta(\mathbf{z}|X)$, parametrized

by $\theta$:

$$X \sim p_{\text{data}}(X) \tag{3}$$
$$\mathbf{z} \sim p_\theta(\mathbf{z}|X). \tag{4}$$

Given that the marginal distribution, $p(\mathbf{z})$, is Gaussian, the GAN continues to be trained to transform a Gaussian distribution to a speech distribution, conditioned on semantic tokens. Here, the one-to-one correspondence between $\mathbf{z}$ and the target waveform, $X$, is re-established, allowing the usage of Mel-spectrogram loss to stabilize training and avert mode collapse. During inference, to generate random speakers, samples can be drawn from the marginal distribution, $p(\mathbf{z})$, or, to encode a specific speaker, the maximum a posteriori estimate of the conditional distribution, $p_\theta(\mathbf{z}|X)$, can be used. We refer to the resulting model as *NFA-VoxGenesis*.

The ancestral sampling procedures described in Eq. 3 and Eq. 4 can be generalized to encompass any encoder capable of yielding a conditional distribution, extending even to discriminative speaker encoders. The essential prerequisite here is the feasibility of sampling from the marginal distribution $p(\mathbf{z})$, a condition not satisfied by discriminative speaker encoders as it does not constrain $p(\mathbf{z})$ during training. To address this, a divergence term can be introduced during discriminative speaker encoders training to ensure that the marginal distribution $p_\theta(\mathbf{z})$ approximates a standard Gaussian distribution:

$$\min_\theta \lambda \mathcal{D}_{KL}(p_\theta(\mathbf{z})||\mathcal{N}(\mathbf{0}, \mathbf{I})), \tag{5}$$

where $\mathcal{D}_{KL}$ is the Kullback–Leibler divergence and $\lambda$ control the strength of divergence relate to other encoder loss. The subscript $\theta$ denotes the dependence of the implicit distribution $p(\mathbf{z})$ on the parameters of the speaker encoder. Since $p_\theta(\mathbf{z})$ is accessible only through ancestral sampling, Eq. 5 is executed by computing the mean and the standard deviation of the speaker embeddings within a mini-batch and subsequently computing the divergence with a standard Gaussian. Incorporating the divergence term during the training phase of the speaker encoders enables compatibility of our framework with any speaker encoders. The VoxGenesis model equipped with a speaker encoder trained via cross-entropy is referred as *CE-VoxGenesis*, and when trained with contrastive loss, it is referred as *CL-VoxGenesis*. Because all encoders are trained with Gaussian divergence, we refer to them as Gaussian-constrained speaker encoders as depicted by Figure 1(b). Table 4 illustrates the different variants of VoxGenesis associated with various speaker encoders.

### 3.2 Interpretable Latent Direction Discovery

GANs are often preferred over denoising diffusion models and flow models (Ho et al., 2020; Kingma & Dhariwal, 2018) due to their semantically meaningful latent space. This characteristics enables manipulations to modify various aspects of the generated object (Härkönen et al., 2020; Voynov & Babenko, 2020b). This feature is particularly invaluable in applications where obtaining attribute labels is challenging. In this section, we illuminate how a straightforward application of Principal Component Analysis (PCA) on intermediate features unveils latent directions instrumental for manipulating speaker characteristics. PCA is a canonical technique designed to identify the predominant variations within the data. Our objective is to apply PCA to latent representations to uncover these significant variations or changes that are interpretable to humans. As discussed in (Härkönen et al., 2020), the isotropic distribution of $p(\mathbf{z})$ tends to be ineffective for highlighting the most distinctive change directions, due to its uniform characteristic in all dimensions. To use PCA effectively, we opt for computing them on the output of the mapping network $M(\mathbf{z})$. We randomly sample $\mathbf{z}$ from a Gaussian distribution and compute the corresponding $\mathbf{w} = M(\mathbf{z})$. Singular Value Decomposition (SVD) is then employed to determine the $N$ bases $\{\mathbf{v}\}_{n=1}^N$. For any given speaker representation $\mathbf{z}$, modifications can be performed by moving $\mathbf{w}$ along the direction outlined by the principal component $\mathbf{v}_n$:

$$\mathbf{w}' = M(\mathbf{z}) + s\mathbf{v}_n, \tag{6}$$

where $s$ represents a shift value. $\mathbf{w}'$ is subsequently fed through the deconvolution layers to synthesize speech spoken by the modified speaker. This procedure is applicable to external speaker representations encoded through encoders like NFA or Gaussian-constrained speaker encoders. Figure 2 illustrates the effect of modifying latent codes along the discovered directions. We find principal directions related to gender characteristics, pitch, tone, and emotion. Notably, inter-speaker variations like gender are reflected in the leading Principal Components (PCs), while more subtle intra-speaker

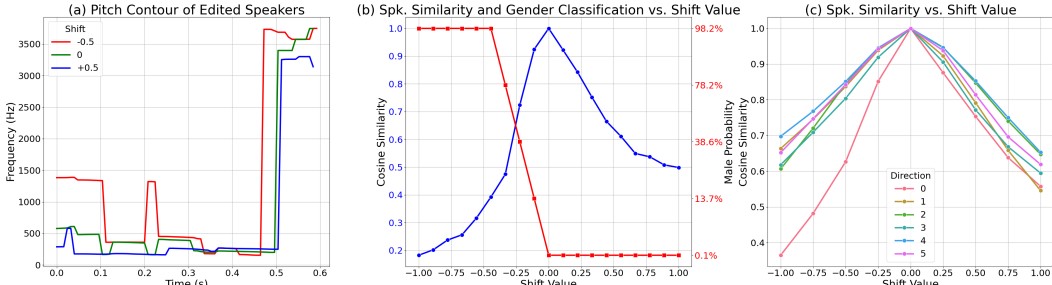

Figure 2: The effect of manipulating different latent directions on speaker similarity, pitch, and gender identification.

nuances like emotion are captured in the latter PCs. As illustrated in Figure 2 (b), manipulating the latent representation of a male speaker along the negative direction of first principal component gradually shifts it towards the sound of a female speaker, as detected by a speech gender classifier. It is noteworthy that between the region of -0.1 and -0.25, speaker similarity remains high (0.9 and 0.7, respectively), and the classifier exhibits ambiguity regarding the speaker's gender. This implies minimal alteration in speaker identity, while rendering the speaker more feminine sounding with subtle modifications. In Figure 2 (a), we demonstrate the effects of modifying the third principal direction, responsible for controlling pitch. The shift in the latent code along this PC (depicted by the green line) apparently lowers the pitch across the entire recording, compared to the original waveform represented by the blue line. Conversely, altering along the opposite direction (illustrated by the red line) elevates the pitch. Figure 2 (c) evaluates the ramifications of shifting different PCs on speaker similarity. It's evident that manipulations employing leading PCs influence speaker similarity more substantially compared to those utilizing later PCs. This phenomenon suggests a potential application of later PCs in refining subtle speaker attributes like emotion and intonation, allowing for nuanced adjustments while preserving the inherent characteristics of the speaker.

### 3.3 VOICE CONVERSION AND MULTI-SPEAKER TTS WITH VOXGENESIS

With the integration of NFA (Lin et al., 2023) or Gaussian-constrained speaker encoders, VoxGenesis can be effectively employed for voice conversion and multi-speaker Text-to-Speech (TTS). Given a speaker reference waveform, denoted as $X_b$, and a content reference waveform, denoted as $X_a$, VoxGenesis enables the conversion of the speaker identity in $X_a$ to that in $X_b$, while the speech content in $X_a$ remains unchanged. This is represented mathematically as:

$$\hat{X}_{a\to b} = G\left(\mathbf{z}_b \mid Y_a\right), \quad \text{where} \quad \mathbf{z}_b = \arg\max_{\mathbf{z}} p_\theta(\mathbf{z} \mid X_b). \quad (7)$$

$Y_a$ represents the semantic token that is extracted from the content reference waveform $X_a$. Essentially, this capability allows for the transformation of speaker identity of the given speech content without altering the content of the speech. We can also sample from a Gaussian distribution to generate a novel speaker speaking the content in $X_a$:

$$\hat{X}_{a\to?} = G\left(\mathbf{z} \mid Y_a\right), \quad \text{where} \quad \mathbf{z} \sim \mathcal{N}(\mathbf{0}, \mathbf{I}). \quad (8)$$

VoxGenesis can also be deployed as a speaker encoder and vocoder for a multi-speaker TTS. For this application, we initially discretize speech features utilizing HuBERT (Hsu et al., 2021) and subsequently train a Tacotron model to predict the discrete tokens.

### 4 EXPERIMENTAL SETUP

#### 4.1 MODEL CONFIGURATION AND TRAINING DETAILS

All VoxGenesis variants and baseline models including TacoSpawn, VITS, and SpeechResynthesis were trained using the train-clean-100 and train-clean-360 split of LibriTTS-R (Zen et al., 2019; Koizumi et al., 2023). Audio files are downsampled to 16kHz to ensure compatibility with the 16kHz

Table 1: Speaker generation evaluation. Subjective metrics results are reported with a 95% confidence interval.

| Method | FID ↓ | Spk. Similarity ↓ | Spk. Diversity ↑ | MOS ↑ |
|---|---|---|---|---|
| Ground Truth | - | 0.22 | 4.22±0.07 | 4.3 ± 0.09 |
| TacoSpawn (Stanton et al., 2022) | 0.18 | 0.59 | 3.85±0.09 | 3.54 ± 0.09 |
| Vanilla-VoxGenesis | 0.17 | 0.38 | 3.96±0.07 | 3.92±0.07 |
| NFA-VoxGenesis | 0.14 | 0.30 | **4.17±0.09** | **4.22±0.06** |
| CL-VoxGenesis | 0.16 | 0.36 | 4.02±0.12 | 3.74±0.08 |
| CE-VoxGenesis | **0.11** | **0.28** | 4.11±0.07 | 4.13±0.09 |

models. For training vanilla-VoxGenesis, NFA-VoxGenesis, and CL-VoxGenesis, we did not use speaker labels or any meta data. For CE-VoxGenesis, we used the speaker labels. We used HuBERT Large as semantic encoder (Ott et al., 2019). Different from (Lin et al., 2023), NFA was trained with an EM algorithm using HuBERT's features and discrete tokens. The embeddings dimension of NFA speak vector is 300. For CE-VoxGenesis and CL-VoxGenesis, the speaker encoders were trained using cross-entropy and contrastive loss on a X-vector network (Snyder et al., 2018), respectively. Because the HuBERT features have a larger time span than the MFCC features used in the original HiFi-GAN, we adjusted the upsample parameters in the transpose convolution layer to [10, 4, 2, 2]. We used the Adam optimizer with a learning rate of 0.0002 and the betas set to 0.8 and 0.99. The training segment length was set to 8,960 frames.

## 4.2 EVALUATION METRICS FOR SPEAKER GENERATION

We used Fréchet Inception distance (FID) (Heusel et al., 2017) on speaker embeddings to compare the generated speaker distribution and the training speaker distribution. We used 50,000 randomly sampled utterances to evaluate the FID score. Because a model that simply memorizes the training speakers would achieve a very low FID score and it is easy to memorize speaker embeddings with speaker labels, which would not align with our goal of novel speaker generation. To complement FID, we used an additional subjective metric that measures the similarity between the generated speakers and the training speakers.

## 5 RESULTS

Given the nature of our work, we believe that it would be more informative for readers to listen to the audio samples for comparisons. The demo page is available at `https://bit.ly/VoxGenesis`.

## 5.1 SPEAKER GENERATION EVALUATION

In this section, we evaluate the diversity, speech quality, and similarity of the generated speakers in comparison to the training speakers. Table 1 presents these evaluations for TacoSpawn (Stanton et al., 2022) and four VoxGenesis variants, with ground truth included for reference. We can see that all four variants of VoxGenesis produce lower FID scores than TacoSpawn. This suggests that VoxGenesis is more effective in capturing the speaker distribution. Moreover, VoxGenesis speaker similarity is also significantly lower than TacoSpawn, suggesting that the generative process relies less on memorization of the training speakers. Among the variants, the unsupervised version of VoxGenesis registers a higher FID score than its supervised counterpart—a foreseeable outcome given its lack of access to speaker labels. Despite no discernible difference in speech quality, Vanilla-VoxGenesis records the lowest diversity score, as reflected by both FID and speaker diversity score, signaling some degree of mode collapse occurring in Vanilla-VoxGenesis training. During the auditory evaluation, we found that VoxGenesis tends to generate speakers who exhibit distinct characteristics and speak with better intonation and emotion, in contrast to the more neutral-toned speakers produced by TacoSpawn. This distinction is reflected in the MOS and speaker diversity scores, where all four VoxGenesis variants outperform TacoSpawn.

## 5.2 Edibility of the Latent Space

In this section, we evaluate the edibility of the VoxGenesis latent space. Specifically, the objectives are to investigate: (1) whether editing impacts the speech quality of the recording, (2) the extent to which editing alters the identity of the speaker, and (3) whether the editing along a direction has consistent effects and generalizes to both the internal latent code and externally encoded speakers. To address these questions, we provided 10 edited sample for each editing direction, and engaged human assessors to evaluate both the speech quality, measured by MOS, and identifiability, measured by the "successful ID rate". Additionally, a pre-trained speaker classifier (Snyder et al., 2018) was employed to assess the similarity among the edited speakers. For the identifiability experiments, assessors were asked to identify the changes induced by editings, given 10 options, which included 4 real directions utilized in the experiment and 6 random distractors. We conducted experiments on both internal representations (samples from Gaussian prior) and external speakers (extracted from test speech files). The outcomes of these experiments are shown in Figure 3, where the first row

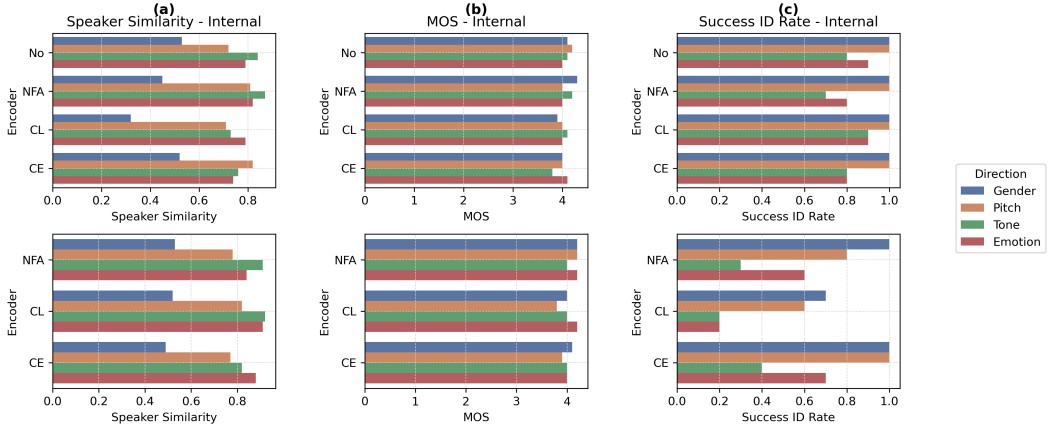

Figure 3: Barplot showing the effect of editing the latent representations on speaker similarity, speech quality, and identifiability.

shows the results of internal speaker editing and the second row show the results of external speaker editing, and each column shows the different aspects of evaluation. In the column (a), it is observed that, with the exception of gender, most edits retain high speaker similarity, indicating that the majority of the latent directions induced are within-speaker changes. In the column (b), which measures speech quality, it is evident that editing does not compromise the quality of the speech much; most MOS of the edited files exceed 4. In the column (c), which examines the identifiability of the editing, it is apparent that changes in internal representation are quite noticeable to the listener, as indicated by the very high successful ID rate in the first row of the (c) column. Nevertheless, we noticed that manipulating external code is notably more challenging than adjusting internal code. This is particularly apparent for more nuanced attributes such as tone and emotion, where the successful ID rate experiences a significant drop between the two rows of the column (c). Although Success ID Rate decreases in all instances, different encoders exhibit distinct behaviors, with NFA and supervised speaker encoder demonstrating more robust editing capabilities.

## 5.3 Zero-shot Voice Conversion and Multi-Speaker TTS Performance

In addition to speaker generation and editing, it is straightforward to apply VoxGenesis to voice conversion and multi-speaker TTS tasks. Given that the majority of VC and TTS systems utilize embeddings from discriminative speaker encoders, exploring the performance of an unsupervised approach like NFA-VoxGenesis, which is trained without using any speaker labels, is quite interesting. Therefore, the focus of this evaluation is primarily on NFA-VoxGenesis, and its performance is compared with the state-of-the-art the voice conversion system, Speech Resynthesis (Polyak et al., 2021), and the state-of-the-art multi-speaker TTS system, VITS (Kim et al., 2021).

For zero-shot voice conversion, we randomly selected 15 speakers from LibriTTS-R (Koizumi et al., 2023) test split. We assessed the capability of the model to retain content and maintain speaker fidelity. This is measured by the Word Error Rate (WER) and Equal Error Rate (EER) using a pretrained ASR (Ravanelli et al., 2021) and an (Snyder et al., 2018) model, respectively, alongside the speech naturalness, measured by MOS. The results are documented in Table 2. As can be seen from Table 2, VoxGenesis and Speech Resynthesis exhibit comparable performance in content preservation, as measured by WER. This is anticipated since both VoxGenesis and Speech Resynthesis employ a HuBERT-based model to extract content information. Regarding speaker fidelity, NFA-VoxGenesis surpasses Speech Resynthesis in terms of EER, indicating that the generative speaker encoder of NFA maintains speaker information more effectively than the discriminative speaker encoder in Speech Resynthesis. Additionally, the overall speech quality of NFA-VoxGenesis is superior to that of Speech Resynthesis, as reflected by the higher MOS score. For multi-speaker TTS, we assess the generated speech with a focus on speaker MOS, where evaluators appraise the similarity between the generated speakers and the ground truth speakers, putting aside other aspects such as content and grammar. Additionally, we employ general MOS to measure the overall quality and naturalness of the speech. As indicated in Table 3, VoxGenesis achieves higher MOS scores in both speaker similarity and naturalness. We observed that VoxGenesis preserves speakers characteristics and intonation better, despite the absence of speaker labels during the training.

| Method | Dataset | WER | EER | MOS |
|--------|---------|-----|-----|-----|
| NFA-VoxGensis | LibriTTS-R | 7.56 | **5.75** | **4.21±0.07** |
| Speech Resynthesis | LibriTTS-R | **7.54** | 6.23 | 3.77±0.08 |
| ControlVC | LibriTTS-R | 7.57 | 5.98 | 3.85±0.06 |
| NFA-VoxGensis | LibriTTS | **6.13** | **4.82** | **4.01±0.07** |
| Speech Resynthesis | LibriTTS | 6.72 | 5.49 | 3.42±0.04 |
| ControlVC | LibriTTS | 6.43 | 5.22 | 3.56±0.03 |
| NFA-VoxGensis | VCTK | **5.68** | **2.83** | **4.32±0.08** |
| Speech Resynthesis | VCTK | 6.15 | 4.17 | 3.58±0.09 |
| ControlVC | VCTK | 6.03 | 3.88 | 3.66±0.07 |

Table 2: Results of Voice Conversion Experiments on LibriTTS-R, Original LibriTTS, and VCTK Datasets. The baselines are ControlVC (Chen & Duan, 2022) and Speech Resynthesis (Polyak et al., 2021).

| Measurement | Dataset | NFA-VoxGenesis | VITS | StyleTTS | FastSpeech2 |
|-------------|---------|----------------|------|----------|-------------|
| Spk. MOS | LibriTTS-R | **4.03±0.09** | 3.63±0.2 | 3.68±0.09 | 3.55±0.12 |
| MOS | LibriTTS-R | **4.15±0.08** | 3.8±0.09 | 3.94±0.07 | 3.82±0.07 |
| Spk. MOS | LibriTTS | **4.05±0.06** | 3.42±0.11 | 3.74±0.06 | 3.77±0.11 |
| MOS | LibriTTS | **4.02±0.08** | 3.54±0.07 | 3.82±0.08 | 3.72±0.08 |
| Spk. MOS | VCTK | **4.3±0.07** | 3.95±0.09 | 4.02±0.07 | 3.81±0.06 |
| MOS | VCTK | **4.42±0.09** | 4.03±0.08 | 4.09±0.06 | 4.18±0.07 |

Table 3: Comparison of Multi-Speaker TTS Performance on LibriTTS-R, Original LibriTTS, and VCTK, Featuring Benchmarks Against VITS (Kim et al., 2021), StyleTTS (Li et al., 2022), and FastSpeech2 (Ren et al., 2020)

# 6 CONCLUSIONS

In this paper, we introduced VoxGenesis, a deep generative model tailored for voice generation and editing. We demonstrated that VoxGenesis is capable of generating realistic speakers with distinct characteristics. It can also uncover significant, human-interpretable speaker variations that are hard to obtain labels. Furthermore, we demonstrated that VoxGenesis is adept at performing zero-shot voice conversion and can be effectively utilized as both a vocoder and a speaker encoder in multi-speaker TTS.

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

# A  APPENDIX

| Model | Input to GAN | Encoder Objective | Speaker Labels | Speaker Generation | Speaker Encoding | Speaker Edit |
|-------|------|------|------|------|------|------|
| Vanilla-VoxGenesis | Noise | NA | No | Yes | No | Internal |
| NFA-VoxGenesis | Emb. | Likelihood | No | Yes | Yes | Any |
| CL-VoxGenesis | Emb. | Contrastive + KL-divergence | No | Yes | Yes | Any |
| CE-VoxGenesis | Emb. | Cross-entropy + KL-divergence | Required | Yes | Yes | Any |

Table 4: Summary of VoxGenesis models with different speaker encoders.

## A.1  ADDITIONAL EXPERIMENTS

We have supplemented the experiments with different SSL models for content modeling in Table 5.

| Speaker Module | Content Model | WER | EER | MOS |
|-------|------|------|------|------|
| NFA | HuBERT | 7.56 | 5.75 | 4.21±0.07 |
| NFA | w2v-BERT | 7.22 | 5.63 | 4.1±0.09 |
| NFA | ContentVec | 7.04 | 5.65 | 4.25±0.05 |

Table 5: The Effect of Using Different SSL Module for Content Modeling

## A.2  ADDITIONAL SYSTEMS DESCRIPTION

The speaker embeddings networks employed in our study, both contrastive and supervised, are based on the x-vector architecture. The supervised x-vector network is trained using a combination of cross-entropy loss and KL-divergence, with the weighting factor $\lambda$ set to 1. In contrast, the contrastive x-vector network is trained using the NT-Xent loss, also with $\lambda$ set to 1. Both networks undergo training on the same dataset as VoxGenesis. Regarding Tacospawn, we implement ancestor sampling, which involves initially sampling from a mixture distribution and then from a Gaussian distribution. For all VoxGenesis models, we directly sample from a standard normal Gaussian distribution.

## A.3  ADDITIONAL DETAILS ABOUT SPEAKER GENERATION EVALUATION

Specifically, we used a pre-trained x-vector network (Snyder et al., 2018) to retrieve the top-3 most similar speech segments, and then asked 20 human evaluators to assess the similarity between the generated speaker and the retrieved ones. We used a three-point scale from 0 to 1 to represent the evaluators' opinions, with 1 being that the retrieved audio is very likely from the generated speaker and 0 being that the retrieved audio is unlikely to be from the generated speaker. We refer to the

score as the speaker similarity score. Additionally, we asked the evaluators to rate the diversity of the generated speakers on a scale from 0 to 5, with 0 indicating no diversity and 5 indicating that every utterance sounded like it was spoken by a different speaker. We refer to this metric as the diversity score. Finally, we asked the evaluators to rate the naturalness of the speech using the standard MOS scale, with an interval of 0.5. We utilized crowd-sourcing for the subjective evaluations.

## A.4 LIBRITTS AND LIBRITTS-R GENERATED SPEAKER QUALITY COMPARISON

| Method | Dataset | FID | Spk. Similarity | Spk. Diversity | MOS |
|--------|---------|-----|-----------------|----------------|-----|
| NFA-VoxGenesis | LibriTTS-R | 0.14 | 0.3 | 4.17±0.09 | 4.22±0.06 |
| NFA-VoxGenesis | LibriTTS | 0.15 | 0.23 | 4.4±0.07 | 4.03±0.05 |

Table 6: LibriTTS and LibriTTS-R Generated Speaker Quality Comparison

