# OpenReview forum: "VoxGenesis: Unsupervised Discovery of Latent Speaker Manifold for Speech Synthesis"
_ICLR.cc/2024/Conference — ICLR 2024 Conference Withdrawn Submission_

### Official Review · Reviewer_cVSz · 2023-10-26

**Soundness:** 3 good
**Presentation:** 3 good
**Contribution:** 2 fair
**Rating:** 5
**Confidence:** 4

**Summary:**

The paper proposes a speaker creation method that uses a pre-trained speaker embedding model together with Gaussian sampling to generate new speakers.

**Strengths:**

1. The proposed model performs better than Tacospawn in generating unseen and diverse speakers.
2. Ablation studies are performed to understand the impact of different design choices.
3. The paper is in general written clearly with a good amount of illustrations and maths.

**Weaknesses:**

1. I think the novelty is somewhat limited. To me, the proposed method is quite similar to a VAE-based speaker encoder but with the encoder changed to a pre-trained speaker embedding module. The mapping network is almost the same as in StyleGAN and is quite similar in principle to AdaSpeech. My main takeaway is that a pre-trained speaker embedding model as the encoder of a VAE works well for generating unseen speakers, and NFA is a promising speaker embedding model compared to other pre-trained speaker embedding model.
2. The baselines compared are quite old by deep learning standards. There are a few follow-up works to TacoSpawn. For zero-shot VC there are many more recent works (e.g. ControlVC). The authors state VITS is the "the state of the art" Multi-Speaker TTS but it is by no means true in 2023.

**Questions:**

For Table 1, it is not clear to me for each baseline, what is the exact procedure and hyper-params involved to sample speaker embeddings. I hope the authors can clarify.

---

> ### Author Response · Authors · 2023-11-17
>
> We would like to thank the reviewer for the clear and constructive feedback. We have addressed the questions as follows:
> > Q1 I think the novelty is somewhat limited. To me, the proposed method is quite similar to a VAE-based speaker encoder but with the encoder changed to a pre-trained speaker embedding module. The mapping network is almost the same as in StyleGAN and is quite similar in principle to AdaSpeech. My main takeaway is that a pre-trained speaker embedding model as the encoder of a VAE works well for generating unseen speakers, and NFA is a promising speaker embedding model compared to other pre-trained speaker embedding model.
>
> **A1** We believe the reviewer missed the main contribution of our paper. If we have to put the takeaway in one sentence. It would be “constraining speaker embedding distribution into a known form allows speaker generation and speaker attribution manipulation”. Our contribution is not in introducing NFA per se or training speaker embedding in a VAE encoder-like fashion. Rather we identify a principled way to do speaker generation, i.e. by constraining speaker embedding distribution to a known form.
> For the mapping network, we mainly used it to transform the speaker embedding from an isotropic Gaussian distribution into a non-isotropic one to allow latent manipulation. StyleGAN does not use it for latent manipulation.
>
> > Q2  The baselines compared are quite old by deep learning standards. There are a few follow-up works to TacoSpawn. For zero-shot VC there are many more recent works (e.g. ControlVC). The authors state VITS is the "the state of the art" Multi-Speaker TTS but it is by no means true in 2023.
>
> **A2** We have expanded our experiments to include more baselines and datasets. Specifically, we have now incorporated FastSpeech2 and StyleTTS, both are recent popular papers, into our comparisons for multi-speaker TTS. Additionally, we have included ControlVC as an additional voice conversion baseline to provide a broader perspective on our model's performance.
>
> **Comparison of Multi-Speaker TTS Performance on LibriTTS-R, Original LibriTTS, and VCTK, Featuring Benchmarks Against VITS, StyleTTS, and FastSpeech**
>
> | Measurement | Dataset   | NFA-VoxGenesis  | VITS         | StyleTTS     | FastSpeech2  |
> |-------------|-----------|-----------------------------------|--------------|--------------|--------------|
> | Spk. MOS    | LibriTTS-R | 4.03±0.09                         | 3.63±0.2     | 3.68±0.09    | 3.55±0.12    |
> | MOS         | LibriTTS-R | 4.15±0.08                         | 3.8±0.09     | 3.94±0.07    | 3.82±0.07    |
> | Spk. MOS    | LibriTTS   | 4.05±0.06                         | 3.42±0.11    | 3.74±0.06    | 3.77±0.11    |
> | MOS         | LibriTTS   | 4.02±0.08                         | 3.54±0.07    | 3.82±0.08    | 3.72±0.08    |
> | Spk. MOS    | VCTK       | 4.3±0.07                          | 3.95±0.09    | 4.02±0.07    | 3.81±0.06    |
> | MOS         | VCTK       | 4.42±0.09                         | 4.03±0.08    | 4.09±0.06    | 4.18±0.07    |
>
> **Results of Voice Conversion Experiments on LibriTTS-R, Original LibriTTS, and VCTK Datasets**
>
> | Method            | Dataset   | WER  | EER  | MOS        |
> |-------------------|-----------|------|------|------------|
> | NFA-VoxGensis     | LibriTTS-R| 7.56 | 5.75 | 4.21±0.07  |
> | Speech Resynthesis| LibriTTS-R| 7.54 | 6.23 | 3.77±0.08  |
> | ControlVC         | LibriTTS-R| 7.57 | 5.98 | 3.85±0.06  |
> | NFA-VoxGensis     | LibriTTS  | 6.13 | 4.82 | 4.01±0.07  |
> | Speech Resynthesis| LibriTTS  | 6.72 | 5.49 | 3.42±0.04  |
> | ControlVC         | LibriTTS  | 6.43 | 5.22 | 3.56±0.03  |
> | NFA-VoxGensis     | VCTK      | 5.68 | 2.83 | 4.32±0.08  |
> | Speech Resynthesis| VCTK      | 6.15 | 4.17 | 3.58±0.09  |
> | ControlVC         | VCTK      | 6.03 | 3.88 | 3.66±0.07  |
>
>
> We have checked the follow-up works to Tacospawn. The most popular one is "Creating new voices using normalizing flows" [1]. After reading the paper in detail, we found that their way to do speaker generation is entirely the same as Tacospawn. Therefore, we believe it is less interesting to include it.  We are happy to include an additional speaker generation baselines if the reviewer has suggestions.
>
> [1] Bilinski, P., Merritt, T., Ezzerg, A., Pokora, K., Cygert, S., Yanagisawa, K., Barra-Chicote, R., Korzekwa, D. (2022) Creating New Voices using Normalizing Flows. Proc. Interspeech 2022

---

> > ### Author Response · Authors · 2023-11-17
> >
> > > Q3 For Table 1, it is not clear to me for each baseline, what is the exact procedure and hyper-params involved to sample speaker embeddings. I hope the authors can clarify.
> >
> > A3: The speaker embeddings networks employed in our study, both contrastive and supervised, are based on the x-vector architecture. The supervised x-vector network is trained using a combination of cross-entropy loss and KL-divergence, with the weighting factor λ set to 1. In contrast, the contrastive x-vector network is trained using the NT-Xent loss [1], also with λ set to 1. Both networks undergo training on the same dataset as VoxGenesis. Regarding Tacospawn, we implement ancestor sampling, which involves initially sampling from a mixture distribution and then from a Gaussian distribution. For all VoxGenesis models, we directly sample from a standard normal Gaussian distribution. These methodological details will be included in the revised version of our paper for clarity.
> >
> > [1] Chen, Ting, et al. "A simple framework for contrastive learning of visual representations." International conference on machine learning. PMLR, 2020.

---

> > ### Comment · Reviewer_cVSz · 2023-11-22
> >
> > Thanks for the addition results. However I am still not convinced about the reply to Q1. The authors said that the main takeaway should be “constraining speaker embedding distribution into a known form allows speaker generation and speaker attribution manipulation”, but isn't this just simply the main takeaway of any VAE? For example, when beta-VAE first came out, the main takeaway was that "constraining the latent distribution allows for disentangled attribute manipulation and image generation." Therefore I still think the novelty is limited.

---

### Official Review · Reviewer_fp4x · 2023-10-30

**Soundness:** 2 fair
**Presentation:** 2 fair
**Contribution:** 2 fair
**Rating:** 5
**Confidence:** 4

**Summary:**

This work proposes a neural factor analysis (NFA)-based speech synthesis model for tts, vc, and voice generation. They simply disentangle a HuBERT representation for controlling the semantic guided voice representations.

**Strengths:**

They utilize a neural factor analysis (NFA) for speech representation disentanglement, and adapt the semantic representation with style conditions. This simple modification could improve the TTS and VC performance by adopting it to speech resynthesis frameworks.

**Weaknesses:**

1.	The authors should have conducted more comparisons to evaluate the model performance. They only compare it with speech resynthesis, and the result is a little incremental.

2.	They utilize a NFA which was proposed in ICML 2023. The contribution is weak.

3.	It would be better if you could compare the self-supervised speech representation model for the robustness of your methods by replacing HuBert with any other SSL models.

4.	Using HuBERT representation may induce a high WER. Replacing it with ContentVec may improve the pronunciation.

5.	The semantic conditioned Transformation is utilized in many works. NANSY++ utilizes a time-varying timbre embedding and this is almost the same with this part.

**Questions:**

1.	The authors cited LibriTTS and LibriTTS-R together. Which dataset do you use to train the models? In my personal experience, using LibriTTS-R decreases the sample diversity. Do you have an experience with this?

2.	According to ICLR policy, when using human subjects such as MOS, you may include the evaluation details in your paper.

---

> ### Author Response · Authors · 2023-11-17
>
> Thanks for the clear and constructive feedback. We have addressed the questions as follows:
> > Q1: The authors should have conducted more comparisons to evaluate the model performance. and the result is a little incremental.
>
> A1: We have expanded our experiments to include more baselines and datasets. Specifically, we have now incorporated FastSpeech2 and StyleTTS into our comparisons for multi-speaker TTS. Additionally, we have included ControlVC as an additional voice conversion baseline to provide a broader perspective on our model's performance.
>
> **Comparison of Multi-Speaker TTS Performance on LibriTTS-R, Original LibriTTS, and VCTK, Featuring Benchmarks Against VITS, StyleTTS, and FastSpeech**
>
> | Measurement | Dataset   | NFA-VoxGenesis  | VITS         | StyleTTS     | FastSpeech2  |
> |-------------|-----------|-----------------------------------|--------------|--------------|--------------|
> | Spk. MOS    | LibriTTS-R | 4.03±0.09                         | 3.63±0.2     | 3.68±0.09    | 3.55±0.12    |
> | MOS         | LibriTTS-R | 4.15±0.08                         | 3.8±0.09     | 3.94±0.07    | 3.82±0.07    |
> | Spk. MOS    | LibriTTS   | 4.05±0.06                         | 3.42±0.11    | 3.74±0.06    | 3.77±0.11    |
> | MOS         | LibriTTS   | 4.02±0.08                         | 3.54±0.07    | 3.82±0.08    | 3.72±0.08    |
> | Spk. MOS    | VCTK       | 4.3±0.07                          | 3.95±0.09    | 4.02±0.07    | 3.81±0.06    |
> | MOS         | VCTK       | 4.42±0.09                         | 4.03±0.08    | 4.09±0.06    | 4.18±0.07    |
>
> **Results of Voice Conversion Experiments on LibriTTS-R, Original LibriTTS, and VCTK Datasets**
>
> | Method            | Dataset   | WER  | EER  | MOS        |
> |-------------------|-----------|------|------|------------|
> | NFA-VoxGensis     | LibriTTS-R| 7.56 | 5.75 | 4.21±0.07  |
> | Speech Resynthesis| LibriTTS-R| 7.54 | 6.23 | 3.77±0.08  |
> | ControlVC         | LibriTTS-R| 7.57 | 5.98 | 3.85±0.06  |
> | NFA-VoxGensis     | LibriTTS  | 6.13 | 4.82 | 4.01±0.07  |
> | Speech Resynthesis| LibriTTS  | 6.72 | 5.49 | 3.42±0.04  |
> | ControlVC         | LibriTTS  | 6.43 | 5.22 | 3.56±0.03  |
> | NFA-VoxGensis     | VCTK      | 5.68 | 2.83 | 4.32±0.08  |
> | Speech Resynthesis| VCTK      | 6.15 | 4.17 | 3.58±0.09  |
> | ControlVC         | VCTK      | 6.03 | 3.88 | 3.66±0.07  |
>
> Performance wise we believe the improvement is no incremental. Comparing with TacoSpawn, our NFA-VoxGenesis MOS improvement is 0.68, which by any criterion is significant improvement. The discrepancy in MOS is also demonstrated in sample quality in the demo. For voice conversion, the MOS margin with SpeechResynthesis is 0.44, which is also quit considerable improvement. The improvement is relatively small only in multi-speaker TTS.
>
> > Q2: They utilize a NFA which was proposed in ICML 2023. The contribution is weak.
>
> A2: Our primary contribution is not the utilization of NFA. We demonstrate that regular speaker embedding networks, as well as unsupervised contrastive trained embedding networks, can be effectively used for novel speaker generation and speaker editing. The crux of our approach is constraining the speaker distribution p(z) into a form that allows for sampling. We introduce a principled method for speaker generation. Additionally, our work contributes to style control, particularly in independently controlling pitch, tone, and emotion through latent space editing. This contrasts with Style Tokens [1], which have attribute entanglement issues and differs from Style-TTS [2]  which rely on external references for emotion control. Our pitch control also stands apart from pitch-specific techniques like ControlVC [3] and Fastspeech2 [4], which use pitch features and a dedicated pitch encoders.
>
> Reference:
>
> [1] Wang, Yuxuan, et al. "Style tokens: Unsupervised style modeling, control and transfer in end-to-end speech synthesis." International conference on machine learning. PMLR, 2018.
>
> [2] Li, Yinghao Aaron, Cong Han, and Nima Mesgarani. "Styletts: A style-based generative model for natural and diverse text-to-speech synthesis." arXiv preprint arXiv:2205.15439 (2022).
>
> [3] Chen, Meiying, and Zhiyao Duan. "ControlVC: Zero-Shot Voice Conversion with Time-Varying Controls on Pitch and Rhythm." arXiv preprint arXiv:2209.11866 (2022).
>
> [4] Ren, Yi, et al. "FastSpeech 2: Fast and High-Quality End-to-End Text to Speech." International Conference on Learning Representations. 2021.

---

> ### Author Response · Authors · 2023-11-17
>
> >Q3 & Q4: It would be better to compare the self-supervised speech representation model for the robustness of your methods by replacing HuBert with other SSL models. Using HuBERT representation may induce a high WER. Replacing it with ContentVec may improve the pronunciation.
>
> A3 & A4: Thanks for the suggestion on SSL module. We have included results from other SSL-based content extractors, such as wav2vec-BERT and Content-Vec. The impact on WER is obvious. We will incorporate these findings in our revised paper.
> | Speaker Module | Content Model | WER  | EER  | MOS        |
> |----------------|---------------|------|------|------------|
> | NFA            | HuBERT        | 7.56 | 5.75 | 4.21±0.07  |
> | NFA            | w2v-BERT      | 7.22 | 5.63 | 4.1±0.09   |
> | NFA            | ContentVec    | 7.04 | 5.65 | 4.25±0.05  |
>
> Q5: The semantic conditioned Transformation is utilized in many works. NANSY++ utilizes a time-varying timbre embedding and this is almost the same with this part.
>
> A5: We would like to clarify that we do not claim the semantic conditioned Transformation as our innovation. Instead, our contribution lies in constraining speaker embeddings p(z) to be easily sampled before transformation.
>
> > Q6: The authors cited LibriTTS and LibriTTS-R together. Which dataset do you use to train the models? In my personal experience, using LibriTTS-R decreases the sample diversity.
>
> We used LibriTTS-R to train our model. Because it is a newly released dataset and based on LibriTTS, we cite both papers in case some readers are not familiar with LibriTTS-R. Our observations align with the reviewer: using the original LibriTTS leads to greater diversity in generated speakers compared to models trained with LibriTTS-R. We will include the results on the original LibriTTS in the appendix.
>
> | Method          | Dataset   | FID  | Spk. Similarity | Spk. Diversity | MOS        |
> |-----------------|-----------|------|-----------------|----------------|------------|
> | NFA-VoxGenesis  | LibriTTS-R| 0.14 | 0.3             | 4.17±0.09      | 4.22±0.06  |
> | NFA-VoxGenesis  | LibriTTS  | 0.15 | 0.23            | 4.4±0.07       | 4.03±0.05  |
>
> > Q7: According to ICLR policy, when using human subjects such as MOS, you may include the evaluation details in your paper.
>
> A7: The evaluation process of the Mean Opinion Score (MOS) is in Section 4.2 of our initial submission. This section comprehensively covers the protocols followed during the human subject evaluation.

---

> > ### Comment · Reviewer_fp4x · 2023-11-22
> > **Thansk for your response**
> >
> > Thanks for your detailed answer. It took some time to reply this response. I've read the paper and response several times.
> >
> > I acknowledge that the result is not a little incremental compared to other models you compared. However, there is no recent model. VITS was published in 2021.
> >
> > The most weakness of this paper is a lack of survey for recent models. In my opinion, the TTS pipeline is similar to SPEAR-TTS which generates a semantic token from text, and then generates an acoustic or waveform from the semantic token.
> >
> > In addition, Tacotron-based semantic token generation may have a problem such as repeating and skipping. It would be better if you could add the experiments for robustness such as WER for TTS results.
> >
> > I will raise a score from 3 to 5. However, I'm still thinking this paper does not meet the bar for the high standard of ICLR2024.

---

> > > ### Author Response · Authors · 2023-11-22
> > >
> > > We respect the reviewer’s opinion and efforts. Some quick remarks:
> > >
> > > > I acknowledge that the result is not a little incremental compared to other models you compared. However, there is no recent model. VITS was published in 2021.
> > > The most weakness of this paper is a lack of survey for recent models. In my opinion, the TTS pipeline is similar to SPEAR-TTS which generates a semantic token from text, and then generates an acoustic or waveform from the semantic token.
> > >
> > > Our TTS pipeline is indeed similar to SPEAR-TTS. However, the text-to-semantic and semantic-to-waveform pipeline is not the contribution or novelty of our paper. We never claim that. Instead, our contribution is to propose a principled way to do speaker generations and speaker editing (style control). Therefore, our main experiment is to compare with Google’s Tacospawn, which is the most up-to-date work we know.
> > >
> > > > In addition, Tacotron-based semantic token generation may have a problem such as repeating and skipping. It would be better if you could add the experiments for robustness such as WER for TTS results.
> > >
> > > Because our work does not focus on solving the semantic token generation, we chose a simple and most popular acoustic model (Tacotron). We would like to ask for the reviewer’s opinion on other acoustic model choices for further work. Much appreciated.

---

> > > > ### Comment · Reviewer_fp4x · 2023-11-22
> > > > **Thanks for your response**
> > > >
> > > > Thanks for your response.
> > > >
> > > > I acknowledge that the main contribution of this paper is not TTS pipeline so I may overstate this issue. However, I think that it is difficult to evaluate the performance of speaker generation and speaker editing so the performance may be judged by TTS or VC results. To be honest, these topic are such minor in speech domain and the speaker generation and controlling are just resulted from training the multi-speaker TTS models.
> > > >
> > > > It would be better if you could define the speaker generation more precisely in terms of prosody and voice and this paper is more fit for a speech venue such as TASLP, Interspeech, and ICASSP.

---

### Official Review · Reviewer_dpEi · 2023-10-31

**Soundness:** 3 good
**Presentation:** 3 good
**Contribution:** 3 good
**Rating:** 5
**Confidence:** 4

**Summary:**

This paper proposes an unsupervised voice generation model called VoxGenesis by transforming Gaussian distribution to speech distribution. The proposed VoxGenesis can discover a latent speaker manifold and meaningful voice editing directions without supervision, and the latent space uncovers human-interpretable directions associated with specific speaker characteristics such as gender attributes, pitch, tone, and emotion, allowing for voice editing by manipulating the latent codes along these identified directions.

**Strengths:**

1. It is interesting to utilize Gaussian distribution transformation for unsupervised voice (speech) synthesis so that the model is able to generate realistic speakers with distinct characteristics like pitch, tone, and emotion.

**Weaknesses:**

1. While the idea is promising, the experimental results seem to be limited. Most of the performances are from the ablation studies. The proposed model should compare the performances with the previous works like SLMGAN [1], StyleTTS [2], and LVC-VC [3]. Moreover, the paper only utilizes one dataset, LibriTTS-R. More extensive experiments on different dataset might be necessary.
2. The paper can be more curated. While it is well written paper, it slightly lacks in structure. Since the idea is interesting enough, I would consider adjusting the rating if the paper is more well structured and the additional experiments are conducted.

[1] Li, Yinghao Aaron, Cong Han, and Nima Mesgarani. "SLMGAN: Exploiting Speech Language Model Representations for Unsupervised Zero-Shot Voice Conversion in GANs." 2023 IEEE Workshop on Applications of Signal Processing to Audio and Acoustics (WASPAA). IEEE, 2023.

[2] Li, Yinghao Aaron, Cong Han, and Nima Mesgarani. "Styletts: A style-based generative model for natural and diverse text-to-speech synthesis." arXiv preprint arXiv:2205.15439 (2022).

[3] Kang, Wonjune, Mark Hasegawa-Johnson, and Deb Roy. "End-to-End Zero-Shot Voice Conversion with Location-Variable Convolutions." Proceedings of the Annual Conference of the International Speech Communication Association, INTERSPEECH. Vol. 2023. 2023.

**Questions:**

Please refer to the weakness section.

---

> ### Author Response · Authors · 2023-11-17
>
> Thanks for the clear and constructive feedback. We have addressed the reviewer's questions as follows:
> > Q1.While the idea is promising, the experimental results seem to be limited. Most of the performances are from the ablation studies.
>
> > Q2. The paper can be more curated. While it is well written paper, it slightly lacks in structure.
>
> A: We have made efforts to improve the structure of our paper by expanding experiments with more datasets and comparison with relevant recent works like StyleTTS [1], ControlVC [2], and FastSpeech2 [3].
>
> **Comparison of Multi-Speaker TTS Performance on LibriTTS-R, Original LibriTTS, and VCTK, Featuring Benchmarks Against VITS, StyleTTS, and FastSpeech**
> | Measurement | Dataset   | NFA-VoxGenesis  | VITS         | StyleTTS     | FastSpeech2  |
> |-------------|-----------|-----------------------------------|--------------|--------------|--------------|
> | Spk. MOS    | LibriTTS-R | 4.03±0.09                         | 3.63±0.2     | 3.68±0.09    | 3.55±0.12    |
> | MOS         | LibriTTS-R | 4.15±0.08                         | 3.8±0.09     | 3.94±0.07    | 3.82±0.07    |
> | Spk. MOS    | LibriTTS   | 4.05±0.06                         | 3.42±0.11    | 3.74±0.06    | 3.77±0.11    |
> | MOS         | LibriTTS   | 4.02±0.08                         | 3.54±0.07    | 3.82±0.08    | 3.72±0.08    |
> | Spk. MOS    | VCTK       | 4.3±0.07                          | 3.95±0.09    | 4.02±0.07    | 3.81±0.06    |
> | MOS         | VCTK       | 4.42±0.09                         | 4.03±0.08    | 4.09±0.06    | 4.18±0.07    |
>
> **Results of Voice Conversion Experiments on LibriTTS-R, Original LibriTTS, and VCTK Datasets**
> | Method            | Dataset   | WER  | EER  | MOS        |
> |-------------------|-----------|------|------|------------|
> | NFA-VoxGensis     | LibriTTS-R| 7.56 | 5.75 | 4.21±0.07  |
> | Speech Resynthesis| LibriTTS-R| 7.54 | 6.23 | 3.77±0.08  |
> | ControlVC         | LibriTTS-R| 7.57 | 5.98 | 3.85±0.06  |
> | NFA-VoxGensis     | LibriTTS  | 6.13 | 4.82 | 4.01±0.07  |
> | Speech Resynthesis| LibriTTS  | 6.72 | 5.49 | 3.42±0.04  |
> | ControlVC         | LibriTTS  | 6.43 | 5.22 | 3.56±0.03  |
> | NFA-VoxGensis     | VCTK      | 5.68 | 2.83 | 4.32±0.08  |
> | Speech Resynthesis| VCTK      | 6.15 | 4.17 | 3.58±0.09  |
> | ControlVC         | VCTK      | 6.03 | 3.88 | 3.66±0.07  |
>
> Reference:
>
> [1] Li, Yinghao Aaron, Cong Han, and Nima Mesgarani. "Styletts: A style-based generative model for natural and diverse text-to-speech synthesis." arXiv preprint arXiv:2205.15439 (2022).
>
> [2] Chen, Meiying, and Zhiyao Duan. "ControlVC: Zero-Shot Voice Conversion with Time-Varying Controls on Pitch and Rhythm." arXiv preprint arXiv:2209.11866 (2022).
>
> [3] Ren, Yi, et al. "FastSpeech 2: Fast and High-Quality End-to-End Text to Speech." International Conference on Learning Representations. 2021.

---

> > ### Comment · Reviewer_dpEi · 2023-11-22
> >
> > Thanks for the response, and I have no further questions. I have read the modified version of manuscript, but still the paper seems to be not well curated and not easily readable. While I will raise few individual scores, I will leave my overall score as it is.

---

### Official Review · Reviewer_dHpK · 2023-11-02

**Soundness:** 1 poor
**Presentation:** 1 poor
**Contribution:** 1 poor
**Rating:** 1
**Confidence:** 4

**Summary:**

This paper discusses on an approach on modeling speaker's voice in speech generation models, as well as its application in voice conversion and multispeaker TTS.

**Strengths:**

Not clear to me.

**Weaknesses:**

* The clarity of the presentation and writing needs improvement. Overall, it's difficult to read this paper. Some basic writing styles, like missing necessary parentheses on citations, is very bothersome for reading. The description of the method seems over complicated than what the method actually is.

* A lot of technical incorrectness. Just a few examples:
   - Sec 3.1: "GAN has been the de facto choice for vocoders." This is a false claim and ignores a large arrays of active and important works in the community. There are other popular vocoders choices like WaveNet, WaveRnn, diffusion-based approaches etc.
   - Sec 3.1: "A notable limitation in these models is ... learn to replicate the voices of the training or reference speakers rather than creating new voices." Another false claim. It improper to state for such an "limitation" because that is not the goal of the task of vocoders.
   - Sec 3.1 "consequently, a conditional GAN is employed to transform a Gaussian distribution, rather than mapping speech features to waveforms". This is another improper comparison as what GAN does is to transfer the conditioning features (e.g. speech features) , rather than plain Gaussian noise, to waveforms.
   - Sec 3.1 "It is crucial, in the absence of Mel-spectrogram loss, for the discriminator to receive semantic tokens; otherwise, the generator could deceive the discriminator with intelligible speech." I don't see the connection.
   - Sec 4.2 says speaker similarity is evaluated in a 3-point scale from 0 to 1, but Table 3 shows speaker similarity as 4.x and 3.x values.

**Questions:**

None.

---

> ### Author Response · Authors · 2023-11-20
>
> > Q1 Sec 3.1: "GAN has been the de facto choice for vocoders." This is a false claim and ignores a large arrays of active and important works in the community. There are other popular vocoders choices like WaveNet, WaveRnn, diffusion-based approaches etc.
>
> A1: We do not ignore the large arrays of active and important works in the community. Actually, we explicitly said so during the literature review section.
>
> > “Given the necessity for vocoders in both VC and TTS to invert the spectrogram, there has been a significant effort to improve them. This has led to the development of autoregressive, flow, GAN, and diffusion-based vocoders (Kong et al., 2020; Oord et al., 2016; Prenger et al., 2019; Chen et al., 2020). “
>
>
> > Q2 Sec 3.1: "A notable limitation in these models is ... learn to replicate the voices of the training or reference speakers rather than creating new voices." Another false claim. It improper to state for such an "limitation" because that is not the goal of the task of vocoders.
>
> A2 We do not say the goal of the vocoder is to create the voice anywhere in our paper. We merely state a fact that the current GAN vocoder tries to replicate the voice of the training speaker instead of creating new voices.
>
>
> > Q3 Sec 3.1 "consequently, a conditional GAN is employed to transform a Gaussian distribution, rather than mapping speech features to waveforms". This is another improper comparison as what GAN does is to transfer the conditioning features (e.g. speech features) , rather than plain Gaussian noise, to waveforms.
>
> A3 The whole novelty of our paper is to transform a known distribution (a Gaussian distribution) into a waveform, which are speaker embeddings. We have a hard time finding what is the improper comparison here.
>
> > Q4  Sec 3.1 "It is crucial, in the absence of Mel-spectrogram loss, for the discriminator to receive semantic tokens; otherwise, the generator could deceive the discriminator with intelligible speech." I don't see the connection.
>
> Q4 Mel-spectrum loss is very important to GAN-based vocoder, without it the network does not receive conditional information, in the absence of it, we provided conditional information in the form of semantic tokens.
>
> > Q5 Sec 4.2 says speaker similarity is evaluated in a 3-point scale from 0 to 1, but Table 3 shows speaker similarity as 4.x and 3.x values.
>
> A5 Table 3 shows speaker MOS instead of speaker similarity. We can not understand why the reviewer thinks it is speaker similarity. Table 3 explicitly writes speaker MOS. Both speaker similarity and Speaker MOS  were explicitly defined in our paper.

---

> > ### Comment · Reviewer_dHpK · 2023-11-22
> >
> > Thanks for the responses from the authors. After reading them, I decide to keep my review feedback and the rating scores.
> >
> > Re A5:
> >
> > My understanding of the 3 subjective metrics described in Sec 4.2 are: speaker similarity is between 0-1; diversity score is between 0-5, naturalness range is not described. All 3 of them fall under the term "MOS", which stands for "mean opinion score". This section doesn't describe a metric named "speaker MOS".
> >
> > Table 3 reports two metrics "Spk MOS" and "MOS", which are not quite clear what they refers to as described in Sec 4.2. However, the text in Sec 5.2 says "As indicated in Table 3, VoxGenesis achieves higher MOS scores in both speaker similarity and naturalness." So clearly to me that there is the discrepancy on the "speaker similarity" score ranges between the description in Sec 4.2 and the numbers reported in Table 3.

---

> > > ### Author Response · Authors · 2023-11-22
> > >
> > > We actually pointed of the definition of  the speaker MOS right before we referred to Table 3
> > > > For multi-speaker TTS, we assess the generated speech with a focus on **speaker MOS**, where evaluators appraise the similar ity between the generated speakers and the ground truth speakers, putting aside other aspects such as content and grammar. Additionally, we employ general MOS to measure the overall quality and naturalness of the speech. As indicated in Table 3, VoxGenesis achieves higher MOS scores ....

---

### Comment · Area_Chair_9XpE · 2023-11-21
**Reminder to reviewers to participate in the author/reviewer discussion**

Dear reviewers, this is a reminder that the author/reviewer discussion period ends November 22.

This discussion is indeed supposed to be a dialog, so please respond to the comments from the authors.

AC

---

### Author Response · Authors · 2023-11-22

Dear Reviewers

We have updated the manuscript. Please take a look.

---

### Author Response · Authors · 2023-11-22

We would like to further elaborate on the novelty and contribution of our paper.
Novelty and Contribution:
1. Our main contribution is not in introducing NFA for speech synthesis but rather in building a framework towards a fully generative model for speech synthesis, which is a departure from the predominant vocoder-centric approach where reference speech is central in speaker modeling. We elucidate the transformation from speaker embedding distributions p(z) to speech distributions P(X) when training the GAN vocoder with speaker embedding. This understanding reveals the limitations of existing frameworks in achieving a truly generative model, primarily due to the unknown nature of speaker embeddings p(z). Our analysis and experiment results both demonstrate approaches like Tacospawn, which attempt to approximate p(z) post-training, are inherently limited. We overcome these limitations by constraining p(z) to known distributions during speaker embedding training. It can either be a neural factor analysis with marginal distribution p(z) known or a vanilla speaker embedding network with Gaussian constrain. Our method thus diverges fundamentally from Tacospawn and offers many new possibilities for speaker generation. In fact, the constraint mechanism can be any distribution measure beyond KL such as maximum mean discrepancy, Jensen-Shannon (JS) Divergence, and Wasserstein Distance. P(z) can also be any distribution as long as we know how to sample from it.
2. Innovative Style Control in Speech Synthesis: Our approach to style control, or 'speaker editing', is a significant leap forward. By implementing PCA-based editing, we enable style control without introducing additional modules during training, illustrating the power of a fully functional latent space. Our style control can control pitch, tone, and emotion independently by simple latent space editing. This is in contrast with Style Tokens [1], which suffer from attribute entanglement style control. Our style control is also reference-free, meaning that for example we do not need a reference speech for emotion control and the emotion control is obtained without supervision or reference., which is also quite different from Style-TTS [2] which used a style encoder to capture emotion from a external emotion speech. Our pitch control also stands apart from pitch-specific techniques like ControlVC [3] and Fastspeech2 [4], which use pitch features and dedicated pitch encoder.

Reference

[1] Wang, Yuxuan, et al. "Style tokens: Unsupervised style modeling, control and transfer in end-to-end speech synthesis." International conference on machine learning. PMLR, 2018.

[2] Li, Yinghao Aaron, Cong Han, and Nima Mesgarani. "Styletts: A style-based generative model for natural and diverse text-to-speech synthesis." arXiv preprint arXiv:2205.15439 (2022).

[3] Chen, Meiying, and Zhiyao Duan. "ControlVC: Zero-Shot Voice Conversion with Time-Varying Controls on Pitch and Rhythm." arXiv preprint arXiv:2209.11866 (2022).

[4] Ren, Yi, et al. "FastSpeech 2: Fast and High-Quality End-to-End Text to Speech." International Conference on Learning Representations. 2021.